# New Function of Cholesterol Oxidation Products Involved in Osteoporosis Pathogenesis

**DOI:** 10.3390/ijms23042020

**Published:** 2022-02-11

**Authors:** Yanting Che, Jingzhi Yang, Fen Tang, Ziheng Wei, Yufan Chao, Na Li, Henghui Li, Si Wu, Xin Dong

**Affiliations:** 1College of Sciences, Shanghai University, Shanghai 200444, China; cheyanting@shu.edu.cn; 2Institute of Translational Medicine, Shanghai University, Shanghai 200444, China; yangjingzhi@shu.edu.cn (J.Y.); lihenghui2020@shu.edu.cn (H.L.); 3School of Medicine, Shanghai University, Shanghai 200444, China; 15601767671@shu.edu.cn (F.T.); chaoyufan@shu.edu.cn (Y.C.); nana_li@shu.edu.cn (N.L.); 4Department of Orthopedics, Shanghai General Hospital, Shanghai Jiao Tong University School of Medicine, Shanghai 201620, China; wotaifu@sjtu.edu.cn; 5Department of Genetics, Stanford University School of Medicine, Stanford, CA 94305, USA

**Keywords:** cholesterol oxidation products (COPs), osteoporosis (OP), 20S-hydroxycholesterol (20S-HC), 22S-hydroxycholesterol (22S-HC), 27-hydroxycholesterol (27-HC)

## Abstract

Osteoporosis (OP) is a systemic bone disease characterized by decreased bone strength, microarchitectural changes in bone tissues, and increased risk of fracture. Its occurrence is closely related to various factors such as aging, genetic factors, living habits, and nutritional deficiencies as well as the disturbance of bone homeostasis. The dysregulation of bone metabolism is regarded as one of the key influencing factors causing OP. Cholesterol oxidation products (COPs) are important compounds in the maintenance of bone metabolic homeostasis by participating in several important biological processes such as the differentiation of mesenchymal stem cells, bone formation in osteoblasts, and bone resorption in osteoclasts. The effects of specific COPs on mesenchymal stem cells are mainly manifested by promoting osteoblast genesis and inhibiting adipocyte genesis. This review aims to elucidate the biological roles of COPs in OP development, starting from the molecular mechanisms of OP, pointing out opportunities and challenges in current research, and providing new ideas and perspectives for further studies of OP pathogenesis.

## 1. Introduction

Osteoporosis (OP) is a multifactorial degenerative disease characterized by decreased bone mass, reduced bone mineral density (BMD), and increased risk of fracture [1]. The early symptoms of OP are not obvious and can be easily overlooked by patients until fractures accidentally occur in the hip, spine, or wrist due to a collision or fall. OP has become a global health issue with aging [2]. Approximately, more than one in ten people over 50 years old worldwide have suffered from OP, and more than half of them have fractured during their lifetime [3,4]. The occurrence of OP is closely related to a variety of factors such as aging, genetic factors, unhealthy lifestyle habits, as well as nutritional deficiencies, among which one of the key factors is the imbalance of bone homeostasis [1,5]. Therefore, it is necessary to systematically understand the process of maintenance and regulation of bone homeostasis for the early diagnosis and prevention of OP. Bone homeostasis is closely related to bone metabolism [6]. Previous studies have shown that along with aging, both calcium loss and hormone metabolism disorders result in disruption of lipid metabolism, which finally leads to disturbance of bone homeostasis and consequent OP [7].

Cholesterol oxidation products (COPs) are a class of lipid oxidation products produced through enzymatic or non-enzymatic mechanism from 27-carbon cholesterol, obtained from diet and cholesterol metabolism. They are biosynthetic intermediates of bile acids, steroid hormones, and 1,25-dihydroxyvitamin D, which are usually present in the circulation of human and animal tissues in either free or bound states [8,9,10,11]. Although the COP contents in the human body are extremely low at picogram to nanogram level, they are reported to play important roles in the regulation of human metabolic homeostasis. For instance, they can act as natural ligands for various receptors such as liver X receptor (LXR), estrogen receptor (ER), and retinoic acid receptor, influencing metabolic activities of life [12,13,14,15,16]. COPs have been found to affect the function of some major organs including the brain, eyes, heart, blood vessels, colon, pancreas, bone, and prostate [12]. COPs have attracted widespread interest from researchers because they participate in the development of age-related diseases such as Alzheimer’s disease, atherosclerosis, and OP [17,18,19]. Studies have shown that 27-hydroxycholesterol (27-HC), one of the major COPs in humans, is an endogenous selective estrogen receptor modulator (SERM) that binds ER competitively with estrogen and regulates bone homeostasis [20]. Increasing 27-HC concentration by pharmacological methods leads to a significant reduction in bone trabeculae and cortical bone and finally results in OP [21]. 22S-hydroxycholesterol (22S-HC) and 20S-hydroxycholesterol (20S-HC) are two specific COPs that affect bone homeostasis by promoting osteogenic differentiation of mesenchymal stem cells and inhibiting their lipogenic differentiation [8,22]. These two specific COPs exhibit significant bone regeneration-promoting effects in a rat model of cranial bone defect [8,22].

In this review, we have discussed the novel functions of COPs involved in OP pathogenesis that are underexplored so far as follows: (1) molecular mechanisms of OP; (2) anabolism of COPs; (3) regulatory roles of COPs in OP; and (4) challenges.

## 2. Molecular Mechanisms of OP

OP is one of the “three major killers” that endanger human health, especially for middle-age and elderly populations, and the incidence of OP is continuously increasing every year [23]. OP can be divided into primary osteoporosis and secondary osteoporosis according to the possible concomitant occurrence of other diseases such as diabetes and chronic pancreatic disease [24,25]. Primary osteoporosis can be further divided into postmenopausal osteoporosis, senile osteoporosis, and idiopathic osteoporosis [24,25]. Primary osteoporosis is a physiological degeneration that inevitably occurs along with aging, while secondary osteoporosis is usually raised by genetic diseases, drugs, and so on [26,27,28]. The occurrence of OP is a complex molecular process, and the pathogenesis of different types of OP is not identical, although they all present a state of loss of bone homeostasis [29].

Osteoclasts and osteoblasts are two cell types involved in the regulation of bone homeostasis. Mature osteoclasts are multinucleated giant cells formed by the fusion of mononuclear macrophages differentiated from myeloid progenitor cells in bone marrow, which remove mineralized bone matrix by producing lysosomal enzymes [30]. NF-KB receptor activator of nuclear factor (RANK), RANK ligand (RANKL), and osteoprotegerin (OPG) compose the RANK–RANKL–OPG signal axis, playing an important role in osteoblast genesis and bone activation [31]. RANK is a member of the tumor necrosis factor (TNF) superfamily of transmembrane proteins, which can bind to the RANKL on the osteoclast precursors in the presence of macrophage colony-stimulating factor to promote osteoclast differentiation, survival, and activation of bone resorption [32,33]. OPG competes with RANKL for binding to RANK and inhibits osteoclast differentiation; therefore, the RANKL/OPG ratio is a key determinant of osteoclast differentiation and bone resorption [34,35].

Osteoblasts are the main functional cells used for bone formation and are responsible for the synthesis, secretion, and mineralization of the bone matrix. Osteoblasts are derived from bone marrow mesenchymal stem cells (BMSCs). However, the differentiation of BMSCs is multidirectional and involved in several different cell systems, including adipocytes, chondrocytes, cardiomyocytes, and fibroblasts, in addition to osteoblasts [7]. The differentiation of BMSCs toward adipocytes but not to osteoblasts is one of the most important causes of dysregulation of bone homeostasis [36]. It is shown that peroxisome proliferator-activated receptor γ (PPAR-γ) and the Wnt signaling pathway play important roles in regulating the lipogenic differentiation and osteogenic differentiation of BMSCs [37]. PPAR-γ is a master regulator of adipogenesis and can promote adipocyte maturation [37,38]. The Wnt signaling pathway plays multiple roles in bone homeostasis, including regulating the differentiation of BMSCs; controlling the survival of mature osteoblasts; and regulating expression of the genes required for bone matrix synthesis and mineralization [38,39]. In addition, the Wnt signaling pathway is also a downstream pathway of the hedgehog (Hh) signaling pathway in BMSCs differentiation [40]. Hh signaling not only promotes osteogenic differentiation of BMSCs by upregulating expression of the osteoblastic differentiation genes such as Runt-related transcription factor 2 (Runx2), bone morphogenic protein (BMP), and growth factors, but it also inhibits adipogenesis by suppressing the expression of lipid-derived transcription factor Gate2 [40,41,42]. Additionally, the Notch signaling pathway, MEK/EPK signaling pathway, and protein kinase C/protein kinase A (PKC/PKA) pathway are also involved in osteoblast differentiation and maturation [43,44].

Osteoblasts and osteoclasts work together to regulate bone homeostasis, communicate with each other through cell–cell interactions, cytokines, and extracellular matrix interactions [45]. The tight interplay between bone formation and bone resorption is essential to the maintenance of bone homeostasis. The balance could be broken while bone formation decreases or bone resorption increases, thus triggering the occurrence of OP [46]. Bone density continues to decrease, bone strength decreases, and bone brittleness increases, resulting in an increased risk of fracture and slow fracture healing [2]. Therefore, to understand the influencing factors of bone homeostasis is crucial for early intervention methods for OP progression.

## 3. Anabolism of COPs

Endogenous cellular cholesterol is one of the risk factors for OP, and its serum level is negatively correlated with BMD [47,48]. Statins is one of the most effective lipid-lowering drugs, which can not only reduce cholesterol content but also directly prevent cholesterol autooxidation and reduce the content of specific endogenous COPs [49]. COPs are important oxidation products in cholesterol metabolism; their structures and synthesis pathways are shown in Figure 1 [50].

### 3.1. The Formation of COPs

#### 3.1.1. Non-Enzymic Oxidation

Non-enzymic oxidation refers to direct attack by reactive oxygen species (ROS) on the free radicals of cholesterol, which occurs mainly on the sterol ring of cholesterol, including 7α-hydroxycholesterol (7α-HC), 7β-hydroxycholesterol (7β-HC), and 7-ketocholesterol (7-KC) [10]. ROS first generates 7α/β-hydroperoxyl-cholesterol by attacking the free radical on the cholesterol 7 carbon; then, 7α/β-hydroperoxyl-cholesterol produces 7α-HC, 7β-HC, and 7-KC [17]. The 7β-HC and 7-KC are interconverted in liver by 11beta-hydroxysteroid dehydrogenase type 1 (11β-HSD1) and 11beta-hydroxysteroid dehydrogenase type 2 (11β-HSD2) [51]. In addition, 7-KC is produced by the oxidation of 7-dehydroxycholesterol (7-DC) that is catalyzed by cholesterol 7α-hydroxylase A1 (CYP7A1) [52]. 7-KC is one of the most common non-enzymatic COPs produced in vivo due to high stability [17]. 5,6α-Epoxy cholesterol (5,6α-EC), 5,6β-epoxy cholesterol (5,6β-EC), and 25-hydroxycholesterol (25-HC) are also produced through non-enzymatic reactions [53].

#### 3.1.2. Enzymic Oxidation

Cytochrome P450 is one of the most reported oxidoreductase enzymes, including CYP7A1, cholesterol 7α-hydroxylase B1 (CYP7B1), cholesterol 27-hydroxylase A1 (CYP27A1), and cholesterol 46-hydroxylase A1 (CYP46A1) [54]. 7α-HC can be produced either by non-enzymatic mechanisms or by the catalytic oxidation reaction of CYP7A1, in which the enzyme-catalyzed oxidation reaction is one of the rate-limiting steps in bile acid synthesis. Therefore, the concentration of serum 7α-HC is used to evaluate the changes of bile acid synthesis rate [55,56]. 7α-HC is further oxidized to 7α-hydroxychosterone (7α-HCone) by dehydrogenase 3β-hydroxysteroid dehydrogenase 7 (HSD3B7) [17]. Similar to 7α-HC, 25-HC is also produced by oxidation catalyzed by cholesterol 3-hydroxylase A4 (CYP3A4), CYP27A1, and 25-hydroxylase (Ch25h). Among them, Ch25h is a type of hydroxylase [57]. CYP27A1 is a non-specific enzyme that is mainly found in liver and extrahepatic tissues, which oxidizes cholesterol to 27-HC and also oxidizes free radicals on carbon 27 in cholesterol oxidation products to hydroxyl groups, for instance, converting 7α-HC to 7α,27-dihydroxycholesterol (7α,27-diHC) [58]. CYP46A1 is responsible for the synthesis of 24-hydroxycholesterol (24-HC) in the brain by converting cholesterol to 24-HC, which further contributes to cholesterol elimination [59]. Then, 24-HC passes the blood–brain barrier and gets into the bloodstream for liver catabolism [60].

24,25-Epoxy cholesterol (24,25-EC) is a specific COPs that is not directly produced by cholesterol oxidation but via CYP46A1-catalyzed oxidation in a shunt branch of the cholesterol synthesis pathway [61,62].

### 3.2. Dietary Sources of COPs

Dietary sources of COPs are mainly derived from foods high in cholesterol, including eggs, animal tissues, and dairy products [63]. The prolonged exposure of these cholesterol-containing diets to air leads to the reactions with oxygen to produce COPs. Prolonged storage as well as treatment with high temperature intensifies this oxidation reaction, leading to an increase in the production of COPs [64,65]. Dietary intake of COPs is absorbed in intestines and transported into circulation within celiac particles [10,66]. Studies have shown that COPs are absorbed differently in the intestine; most COPs are more easily absorbed as compared to cholesterol [10,60,67].

### 3.3. Metabolism of COPs

COP metabolism in humans can be divided into two main metabolic pathways; one is to produce highly polar water-soluble products that are excreted through the metabolic cycle [57,68]. For example, 7α-HC, 25-HC, and 27-HC can be metabolized into bile acid by CY27A1 and CYP7B1 [56,69,70,71]. The 7-hydroxylation and 27-hydroxylation of COPs by CYP27A1 and CYP7B1 are key regulatory steps in the bile acid synthesis pathway [72]. Another metabolic pathway is that free COPs are converted into long-chain fatty acid acyl esters by esterification and then accumulate in tissues [65]. The esterified form of COPs accounts for about 40–90% of the total COPs in the human environment [73]. Since COPs are cytotoxic, it is suggested that esterification could be metabolism protective [59,65].

## 4. Biological Functions of COPs Involved in OP

COPs are important bone homeostasis regulators by regulating important signaling pathways including Hh, Wnt, and RANKL–RANK–OPG [8,67,74]. Although different COPs bear similar structures, their biological roles involved in bone homeostasis regulation are not the same, and some even exert opposite functions; for example, 20S-HC inhibits OP by inducing the osteogenic differentiation of BMSCs, while 27-HC is an SERM that is negatively associated with BMD [75].

### 4.1. 20S-HC and 22S-HC

20S-HC is a specific bone homeostasis regulator that regulates bone homeostasis through Notch, MEK/EPK, and Hh signaling pathways as well as LXR signaling [74,76,77,78]. As shown in Figure 2, 20S-HC can promote osteogenic differentiation and inhibit lipogenic differentiation of BMSCs by directly activating the Hh signaling pathway or by inducing the expression of Notch target genes HES-1 and HEY-1 [76,77,78,79]. A related study reported that 20S-HC also activates LXR to inhibit PPAR-γ and sterol regulatory element binding protein-1c, thereby promoting osteogenic differentiation of BMSCs and inhibiting their lipogenic differentiation [80]. However, 20S-HC in the C3H10T1/2 cell system does not inhibit PPAR-γ expression and adipocyte formation; this contradictory result suggests the existence of different cellular phenotypes of COPs present in different cellular systems [81]. In addition, the anti-fat effect of 20S-HC is closely related to the EPK pathway [8].

Although cellular experiments showed that 20S-HC promotes osteoblast differentiation in BMSCs, 20S-HC alone is not sufficient to induce bone mineralization [8]. The combination of 20S-HC and 22S-HC (SS) not only induces osteogenic differentiation in BMSCs but also blocks and reverses the inhibitory effects of oxidative stress on osteogenic differentiation and induces bone mineralization [8,82]. SS activates Hh signaling by activating the upstream signal factor Smo [83]. The activated Hh signaling on one hand promotes bone mineralization by directly regulating the expression of alkaline phosphatase (ALP), osteocalcin (OCN), and other factors; on the other hand, it increases mitochondrial activity and biogenesis and activates the Wnt signaling pathway, thus stimulating the osteogenic differentiation of BMSCs [74,84,85]. Meanwhile, the expression of Runx2 induced by the Hh signaling pathway in the SS intervention cell systems is completely and slightly inhibited by PKC inhibitors and PKA inhibitors, respectively [80]. This suggested that the basal activity of PKC plays an important role in the SS-induced Hh signaling pathway [83]. In contrast, the osteogenic differentiation of BMSCs involving PKA may be independent of the Hh signaling and directly targets the downstream effectors, mediating osteoblast differentiation [83,85]. Additionally, SS promotes osteogenic differentiation by activating LXR. When Smo inhibitor is added, the increased level of LXR protein is inhibited, suggesting that LXR and Hh signaling interact with each other in promoting the bone differentiation of SS, while the mechanism has not been completely elucidated [74].

SS has synergistic effects with BMP2 in promoting osteoblast differentiation and both promote bone regeneration [8]. Compared with BMP2, SS induces less inflammatory responses in osteoblast differentiation, which may be related to the involvement of the Hh signaling pathway [86]. It was reported that SS induced bone formation in a rat calvaria defect model in vivo [22].

### 4.2. 27-HC

27-HC is a negative regulator of bone homeostasis that is mainly related to ER [75]. ER, as one of the nuclear receptors, is crucial in bone metabolism regulation and induces the expression of RANKL through the formation of a complex dimer by binding to estrogen [87,88]. SERM is a class of compounds structurally similar to estrogen that act on the ER to systematically modulate the effects of estrogen [89,90].

27-HC was first identified as an endogenous SERM that negatively regulates bone homeostasis in vivo [20,91]. Experimental cellular studies showed an increased differentiation of osteoclasts cultured in the lung adenocarcinoma environment by the addition of 27-HC [92]. In animal experiments, the increased levels of 27-HC by injection or gene knockdown resulted in reduced bone density in trabecular and cortical bones [93]. This phenomenon was partially reversed after supplementing estrogen, suggesting that 27-HC negative feedback on bone homeostasis regulation depends on ER [93]. It was observed that 27-HC increased the osteoblast TNF-α and RANKL expressions while it decreased osteoblast differentiation through the activation of LXR in vivo by a systematic examination of LXR targets in bone [21]. 27-HC mediated by LXR in bone is inhibited by dimerization of the complex formed by 27-HC and ER [21]. 27-HC decreased osteoblast differentiation and increased osteoclastogenesis through the combined action of ER and LXR, leading to increased bone resorption in mice [21].

A nested case-cohort study in the Women’s Health Initiative found that 27-HC levels alone were not associated with fracture risk in postmenopausal women, whereas the ratio of 27-HC to bioactive estradiol concentrations was found to be useful in the assessment of fracture risk [94]. Taken together, the amount of 27-HC and estrogen is required to maintain a normal bone phenotype. Once 27-HC concentration is elevated in vivo, it provides negative feedback to bone homeostasis through LXR and ER.

### 4.3. Other Cholesterol Oxidation Products

25-HC, similar to 22S-HC, promotes osteoclastogenesis by inducing RANKL expression and inhibits lipogenic differentiation of BMSCs [67]. The increase in PPAR-γ mRNA expression in DMITro-treated C3H10T1/2 cell system was inhibited by 25-HC, which had no significant decrease after the addition of Hh signaling inhibitors or LXR activators [81]. This indicates that 25-HC has a significant inhibitory effect on the lipogenic differentiation of C3H10T1/2 cells, and its inhibitory effect does not significantly correlate with the Hh signaling and LXR [81].

7α,25-dihydroxycholesterol (7α,25-diHC) is a downstream product of 25-HC and can be secreted in large quantities by osteoblast precursor cells [95]. The secretion of 7a,25-diHC by osteoblast precursor cells elevates the secretion of EBI2 signaling, which promotes the migration of osteoblast precursors to the bone surface and decreases osteoblast differentiation [95].

7-KC is the most prevalent COPs produced by non-enzymatic mechanisms in vivo. The endogenous 7-KC causes cellular damage through a variety of stress response pathways that are closely associated with a variety of age-related diseases [18]. 7-KC promotes autophagy in the BMSCs of acute myeloid leukemia patients by increasing the production of ROS in BMSCs [96]. MC3T3-E1 cell experiments also demonstrated that 7-KC-induced apoptosis was closely associated with increased ROS production and enhanced oxidative stress in endoplasmic reticulum [97]. It was reported that 7-KC directly oxidized the intracellular transcription factor EB in osteoblasts by increasing the content of reactive oxygen species in osteoclasts, enhanced the nuclear translocation of transcription factor EB, and induced cellular autophagy, which in turn led to an increase in the number and activity of osteoblasts and disturbance in bone homeostasis [98].

Cholestane-3β,5α,6β-triol (C-triol), one of the most vascularly toxic COPs in vivo, is significantly elevated in the plasma of patients with Niemann–Pick type C1 [99,100]. The effect of C-triol on bone was manifested in two aspects: inhibition of osteogenic differentiation of BMSCs and promotion of apoptosis of BMSCs. C-triol inhibits osteoblast differentiation and matrix mineralization by the inhibition of ALP activity and OCN expression [101]. C-triol promotes the apoptosis of BMSCs that is associated with the increased intracellular calcium and calcium-dependent ROS production [101].

## 5. Research Challenges

With the biological importance in cholesterol metabolism, preliminary progress has been made in the study of COPs affecting the development of OP by regulating bone homeostasis. The roles of COPs involved in OP are mainly elucidated based on cell experiments, whereas the evidence in animal or clinical studies is rarely being reported. Therefore, in vivo studies should be further conducted to understand the underlying mechanism of COPs in OP. Based on the reported literature, there is a limited number of research of OP in either animal or human probably due to the following reasons.

### 5.1. COPs Standard Compounds

As the structures of COPs showed strong similarities and most of them are isomers, standard substances are critical to be introduced in distinguishing these isoforms for individual COPs analysis.

Nevertheless, the majority of the standard compounds of COPs are not available in current commercial or industrial markets; only a few of them could be found from seldom chemical suppliers with very high expense. These issues bring the difficulties for researchers to start a comprehensive study with OP-related COPs.

Even though the readily standard substances are not yet for a large-scale production, several research laboratories provided synthesis methods for COPs: (1) Synthesis from natural products such as saponins; (2) Chemical synthesis from cholesterol; (3) Catalytic oxidation of cholesterol through electrochemical reaction [102,103,104,105,106].

However, these synthesis methods are mainly focused on hydroxycholesterol with a limited variety so far. In addition, these synthesis methods usually have disadvantages of complicated protocols and with low yields [102,103,104]. Furthermore, it cannot avoid the interference raised by self-oxidation of cholesterol when using cholesterol as a raw material for synthesis [105].

To understand the underlying correlations and functions of COPs in OP, collecting as many as possible standard substances of COPs is in need for comprehensive methodology development in the future. This will not only help us to observe the behaviors of COPs in living systems but also bring us opportunities to explore the complicated pathophysiology of OP.

### 5.2. COPs Extraction in Biological Sample

COPs are low abundant compounds and have susceptibility to be interfered by cholesterol oxidation in study [105]. Appropriate preparation and analytical methods are crucial for COPs extraction in various biological samples [107]. Currently, there are three main methods of sample preparation for COPs: protein precipitation, liquid–liquid extraction, and solid-phase extraction. These commonly applied methods either extract large amounts of lipids and other irrelevant metabolites that greatly interfere with the detection of COPs or cause a loss during the analyte’s purification. Karuna et al. found that the accurate quantification of 7α,25-diHC and 7α,27-diHC could be achieved by processing plasma samples through slow precipitation protein with cold ethanol, while a low-temperature environment is required for the stability of extracts [108]. The Folch and Bligh and Dyer extraction is widely used to extract lipids by using a mixture of chloroform and methanol as lipid extractant, while the volatility of this organic mixture is not suitable for a high-throughput preparation and MS measurements [109,110,111,112]. In addition, hexane, isopropanol, and ethyl acetate present more similar polarities to COPs than chloroform and are also commonly used for the extraction of COPs [113,114,115]. However, an unsuitable type or amount of extraction solvents may cause the loss of COPs during the extraction process and affect the subsequent determination results. Solid-phase extraction (SPE) purifies standards by using selective adsorption and selective chromatographic separation principles. Using such a filter membrane will effectively separate the measured substances from interfering components and enhance the detection ability of analytes, while the SPE usually filters out impurities and analytes together that lead to a reduction after the purification [116,117,118]. Even current extraction methods have provided different solutions; a more specific extraction approach will enable a robust and accurate determination of COPs in human specimen. Special in the future application, a specific and simplified preparation protocol will save a lot of time for laboratory diagnosis in clinics.

### 5.3. Targeted Quantitation Approach

The quantities of COPs in vivo may reflect the pathophysiological process of OP. A robust and accurate quantitation method is necessary to be applied for understanding the abundance of COPs in a living system. Mass spectrometry is a powerful tool for the determination of small compounds in both quantitative and qualitative manners.

More and more laboratories have developed new analytical methods based on GC-MS and LC-MS to quantify the concentration of COPs in biological sample. Dzeletovic et al. established a “gold standard” for the quantification of COPs using GC-MS, while the higher temperature vaporization of gasification changed the structure of some COPs, leading to the limitations of its application [119,120,121,122,123,124]. LC-MS achieves precise quantification through derivatizing COPs to pyridine, nicotine, N, N-dimethylglycine esters, and oximes, while its operation is complicated and inevitable to introduce interference during the measurements [63,99,125,126,127,128,129,130,131,132,133]. Griffiths et al. worked on enzyme-catalyzed oxidative derivatization for the determination of COPs and established a complete set of sample preparation and quantification methods by using enzyme-catalyzed oxidative derivatization combined with micro-liquid extraction for surface analysis and LC-MS; it greatly increased the number of COPs, which can be determination. However, the reaction requires cholesterol oxidase, which has high requirements for experimental conditions and laboratory operators [134,135,136,137]. Currently, the existing methods enable quantifying eight to 10 common COPs with significant differences in different specimens in several studies (Table 1).

An international survey also found the quantification of calibration solutions or inappropriate choice of internal standards led to large variations between the results from different laboratories [138,139]. It can be seen that using exact right standard compounds is critical for the accurate quantitation with mass spectrometry. Recently, a novel idea was provided by Xu’s group that used an alignment algorithm to calculate the theoretical retention time of analytes and further achieve a relative quantification [140]. Applying such methods will help us to solve the difficulties in collecting COPs standard compounds and also save the expense of purchase. Nevertheless, to support this idea, plenty of validation work in the wet lab should be conducted to verify the results derived from in-silicon-theory-based approaches.

## 6. Conclusions and Perspectives

In summary, COPs are involved in the regulation of bone homeostasis through the Hh, RANKL–RANK–OPG, and LXR signaling pathways together. Although different COPs exhibit similar structures, they have different regulatory effects on bone homeostasis, either promote osteogenic differentiation, or inhibit osteogenic differentiation. However, the changes of COPs in OP pathology have not been completely investigated, which limits the application of COPs in clinical diagnosis and targeted therapy. This may be due to the limited number of chemical standards of COPs and the lack of accurate quantitative methods to explore endogenous COP concentrations. Therefore, the following suggestions are proposed: (1) optimizing synthesis methods of COPs chemical standards would help to reduce the market cost as well as to increase the possibility of the large-scale commercialization of COPs chemical standards; (2) developing a simple and rapid method to measure COPs with high coverage, sensitivity, and accuracy; (3) developing a new algorithm-based quantitation method that only requires the retention time and compound polarity to help us to figure out the issue of lacking chemical standards, which may need a lot of validation work to prove its feasibility.

## Figures and Tables

**Figure 1 ijms-23-02020-f001:**
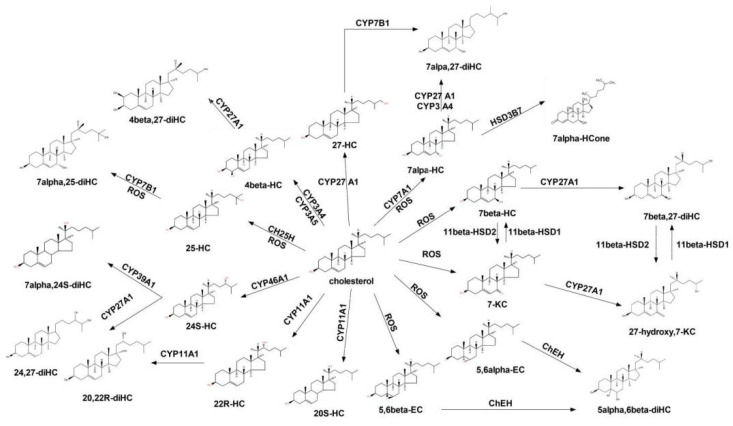
Chemical structure and pathway of COPs. The rectangle is the COPs structure; the ellipses are the enzymes involved in the reaction. 4beta-HC: 4β-hydroxycholesterol; 27-HC: 27-hydroxycholesterol; 7alpha-HC: 7α-hydroxycholesterol; 7alpha,27-diHC: 7α,27-dihydroxycholesterol; 7alpha-HCone: 7α-hydroxychosterone; 7beta-HC: 7β-hydroxycholesterol; 7beta,27-diHC: 7β,27-dihydroxycholesterol; 7-KC: 7-ketocholesterol; 26-hydroxy,7-KC: 26-hydroxy,7-ketocholesterol; 5,6alpha-EC: 5,6α-epoxy cholesterol; 5,6beta-EC: 5,6β-epoxy cholesterol; 20S-HC: 20S-hydroxycholesterol; 22R-HC: 22R-hydroxycholesterol; 20,22R-diHC: 20,22R-dihydroxycholesterol; 24S-HC: 24S-hydroxycholesterol; 24,27-diHC: 24,27-dihydroxycholesterol; 7alpha,24S-diHC: 7α,24S-dihydroxycholesterol; 25-HC: 25-hydroxycholesterol; 7alpha,25-diHC: 7α,25-dihydroxycholesterol; 4beta-HC: 4β-hydroxycholesterol; 4beta,27-diHC: 4β,27-dihydroxycholesterol; ROS: reactive oxygen species; CYP27A1: cholesterol 27-hydroxylase A1; CYP7B1: cholesterol 7α-hydroxylase B1; CYP3A4: cholesterol 3-hydroxylase A4; HSD3B7: dehydrogenase 3β-hydroxysteroid dehydrogenase 7; CYP7A1: cholesterol 7α-hydroxylase A1; 11β-HSD1: 11beta-hydroxysteroid dehydrogenase type 1; 11β-HSD2: 11beta-hydroxysteroid dehydrogenase type 2; CH25H: 25-hydroxylase; CYP11A1: cholesterol 11-hydroxylase A1; CYP3A4: cholesterol 3-hydroxylase A4; CYP3A5: cholesterol 3-hydroxylase A5; CYP46A1: cholesterol 46-hydroxylase A1.

**Figure 2 ijms-23-02020-f002:**
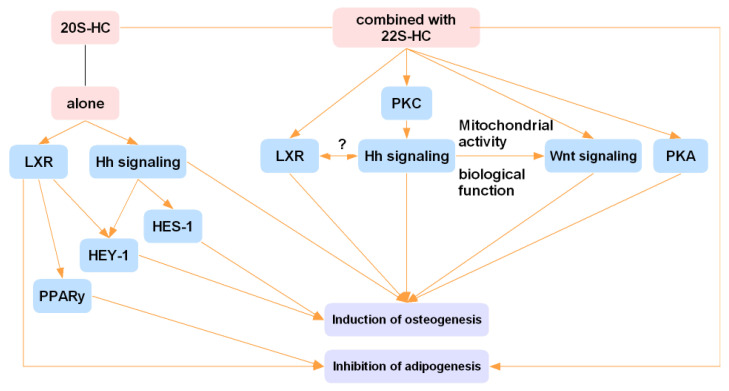
20S-HC and 22S-HC regulate bone homeostasis. LXR: liver X receptor; Hh: Hedgehog; PKC: protein kinase C; PKA: protein kinase A; PPAR-γ: peroxisome proliferator-activated receptor γ.

**Table 1 ijms-23-02020-t001:** The preparation method, determination method, and the content of COPs in human blood.

NO.	Analytical Aspects	Preparatiom Aspects	Analytes	Content	Population Sex	Population Age (Year)	Ref
1	HPLC-ESI-MS	PP Derivatization to N, N-dimethylglycine ester	7-KC ^a^	0.050 ± 0.003 (µmol/L)	-	43.6 ± 9.6	[127]
			C-triol ^a^	0.033 ± 0.004 (µmol/L)			
2	HPLC-APCI-MS	LLE	4β-HC ^a^	30.4 ± 19.9 ng/mL	-	22–27	[115]
3	HPLC-ESI-MS	LLE Derivatization to picolinyl ester	4β-HC ^a^	77 ± 40 ng/mL	-	-	[132]
			7α-HC ^a^	145 ± 82 ng/mL			
			22R-HC ^a^	10 ± 18 ng/mL			
			24S-HC ^a^	51 ± 12 ng/mL			
			25-HC ^a^	31 ± 11 ng/mL			
			27-HC ^a^	117 ± 35 ng/mL			
			24S,25-EC ^a^	2 ± 2 ng/mL			
4	HPLC-ESI-MS	LLE Derivatization to N, N-dimethylglycine ester	24-HC ^a^	38.1 ng/mL	-	-	[128]
			25-HC ^a^	29.7 ng/mL			
			27-HC ^a^	74.1 ng/mL			
			4β-HC ^a^	17.8 ng/mL			
			7α-HC ^a^	29.3 ng/mL			
			7β-HC ^a^	6.9 ng/mL			
			7-KC ^a^	18.8 ng/mL			
5	HPLC-ESI-MS/MS	PP Derivatization to N, N-dimethylglycine ester	7-KC ^b^	3–100 ng/mL	-	0–18	[129]
			C-triol ^b^	3–60 ng/mL			
6	UFLC-ESI-MS	LLE Derivatization to picolinyl ester	4β-HC ^a^	59.0 ± 3.79 ng/mL	female	-	[133]
				43.7 ± 2.31 ng/mL	male	-	
7	GC-MS/MS	LLE Derivatization to trimethylsilyl esters	7α-HC ^a^	0.1 µmol/L	-	19–48	[124]
			7β-HC ^a^	0.1 µmol/L			
			7β-HC ^a^	0.1 µmol/L			
8	UPLC-ESI-MS/MS	LLE Derivatization to picolinyl ester	24S-HC ^a^	65.7 ± 60.6 nmol/L	-	29 ± 10	[134]
			25-HC ^a^	15.3 ± 17.5 nmol/L			
			27-HC ^a^	139.3 ± 76.3 nmol/L			
9	GC-MS/MS	SPE Derivatization to trimethylsiyl ethers	24S-HC ^a^	61.9 ± 14.1 ng/mL	female	-	[125]
			25-HC ^a^	6.4 ±2.2 ng/mL			
			27-HC ^a^	0.14 ± 0.03 µg/mL			
			7α-HCone ^a^	37.6 ± 27.8 nmol/mL			
			24S-HC ^a^	61.3 ± 12.6 ng/mL	male		
			25-HC ^a^	7.5 ± 2.6 ng/mL			
			27-HC ^a^	0.2 ± 0.05 µg/mL			
			7α-HCone ^a^	68.3 ± 92.9 nmol/mL			
10	HPLC-APCI-MS/MS	LLE	7-HC ^b^	1.7–3.3 mg/L	-	average age 34	[117]
			7-KC ^b^	3.0–6.9 mg/L			
			5,6α-EC ^b^	0.9–1.9 mg/L			
			5,6β-EC ^b^	3.5 ± 0.3 mg/mL			
11	LC-ESI-MS/MS	PP Derivatization to N, N-dimethylglycine ester	7-KC ^b^	18.33 ± 3.76 ng/mL	-	under the age of 40	[130]
			C-triol ^b^	9.39 ± 3.17 ng/mL			
			7-KC ^b^	19.72 ± 2.47 ng/mL	-	over the age of 40	
			C-triol ^b^	10.62 ± 2.77 ng/mL			
12	UFLC-ESI-MS	LLE Derivatization to picolinyl ester	24S-HC ^a^	64.4 ± 1.8 ng/mL	-	average age 52.3	[135]
			25-HC ^a^	14.9 ± 1.1 ng/mL			
			27-HC ^a^	139.0 ± 4.7 ng/mL			
			7α-HC ^a^	136.5 ± 12 ng/mL			
			4β-HC ^a^	51.9 ± 2.4 ng/mL			
13	HPLC-APCI-MS	SPE	24-HC ^a^	67 ng/mL	-	average age 39	[118]
			25-HC ^a^	12 ng/mL			
			7α-HC ^a^	55 ng/ml			
			27-HC ^a^	355 ng/mL			
			7-KC ^a^	11 ng/mL			
			27-HC ^b^	57 ng/mL			
14	GC-MS/MS	LLE Derivatization to trimethylsilyl esters	24S-HC ^a^	60.30 ± 14.24 ng/mL	-	68.22 ± 9.17	[123]
			25-HC ^a^	8.52 ± 2.58 ng/mL			

^a^, Content of all COPs containing free and esterified COPs obtaining by an alkaline hydrolysis step; ^b^ Content of free COPs; unreported. ESI: electrospray ionization; APCI: atmospheric pressure chemical ionization; PP: precipitated protein; LLE: liquid–liquid extraction; SPE: solid-phase extraction; 7-KC: 7-ketocholesterol; C-triol: cholestane-3β,5α,6β-triol; 4β-HC: 4β-hydroxycholesterol; 7α-HC: 7α-hydroxycholesterol; 22R-HC: 22R-hydroxycholesterol; 24S-HC: 24S-hydroxycholesterol; 25-HC: 25-hydroxycholesterol; 27-HC: 27-hydroxycholesterol; 24S,25-EC: 24S,25-epoxy cholesterol; 24-HC: 24-hydroxycholesterol; 7β-HC: 7β-hydroxycholesterol; 7α-HCone: 7α-hydroxychosterone.

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
