# Peer review of "New Function of Cholesterol Oxidation Products Involved in Osteoporosis Pathogenesis"

_ijms, 2022, doi:10.3390/ijms23042020_

Round 1

Reviewer 1 Report

The paper “New function of cholesterol oxidation products involved in osteoporosis pathogenesis” is focused on an interesting topic. It is well written and covers some ideas for further researchs.

I suggest to mention, if any, the effects of drugs (e.g. statins) modulating cholesterol levels on cholesterol oxidation products and the possible related association with bone mineral density and fractures; and to mention the effects on vitamin D metabolism.

Author Response

Response:

Thank you for your suggestions.

The effects of cholesterol oxidation products are biosynthetic intermediates of 1, 25-dihydroxyvitamin D, which had been added in Line 50-52 in Page 2, while the connection between cholesterol oxidation products, vitamin D and osteoporosis has been not reported so far.

The association of bone mineral density and fractures has been described in Line 128-130 in Page 3 as follows: “Osteoporosis is a multifactorial degenerative disease characterized by decreased bone mass, reduced bone mineral density, and increased risk of fracture. A continuous decrease in bone density leads to an increased risk of fracture”.

As suggested by reviewer, we have added “the effects of drugs (e.g., statins) modulating cholesterol levels on cholesterol oxidation products” in Line 135-138 on Page 3 as follows: “Statins plays an improtant role in choleaterol metabolism, which can not only reduce cholesterol content, but also directly prevent cholesterol autooxidation and reduce the content of specific endogenous COPs”.

Reviewer 2 Report

The review touches upon a very topical issue of low mineral density. Osteoporosis as a manifestation of comorbidity in a number of diseases is of great importance for elderly patients. The authors have done a lot of work to systematize the available knowledge in the field of the pathogenesis of osteoporosis in terms of the influence of a less studied mechanism of cholesterol oxidation products. The conclusions of the work are justified. Literary sources are used adequately. The article is well prepared, the style of presentation is good. There is a clear logical sequence for presenting the review data.

Author Response

Thank you for your considerations of our review and hope to hear the positive news from you soon.